# Study on the Flavor Compounds of Fo Tiao Qiang under Different Thawing Methods Based on GC–IMS and Electronic Tongue Technology

**DOI:** 10.3390/foods11091330

**Published:** 2022-05-03

**Authors:** Ruirong Lin, Hongfei Yuan, Changrong Wang, Qingyu Yang, Zebin Guo

**Affiliations:** 1Engineering Research Centre of Fujian-Taiwan Special Marine Food Processing and Nutrition, Ministry of Education, Fuzhou 350002, China; 18750156585@163.com (R.L.); yhf199601@163.com (H.Y.); wcr940970864@163.com (C.W.); qingyu980206@163.com (Q.Y.); 2State Key Laboratory of Food Safety Technology for Meat Products, Xiamen 361100, China; 3College of Food Science, Fujian Agriculture and Forestry University, Fuzhou 350002, China

**Keywords:** Fo Tiao Qiang, thawing method, flavor substances, electronic tongue, GC–IMS (gas chromatography coupled to an ion mobility spectrometry)

## Abstract

“Fo Tiao Qiang” is a famous dish with Chinese characteristics. It is delicious, rich in materials, and high in nutritional value. Through physical and chemical analysis, electronic tongue, gas chromatography–ion mobility spectroscopy, and other technologies, the present study explored the quality characteristics and flavor differences of Fo Tiao Qiang by using different thawing methods (natural thawing, ultrasonic thawing, microwave thawing, and water bath thawing). The results show that the protein content was slightly higher in Fo Tiao Qiang with ultrasonic thawing than others. The fat content of the microwave-thawed Fo Tiao Qiang was significantly lower than the other three kinds of samples. After ultrasonic thawing, the number of free amino acids in the samples were the highest and the umami taste was the best. Compared with natural thawing, most of the flavor substances decreased in ultrasonic thawing, microwave thawing, and water bath thawing. However, several substances increased, such as alpha-terpineol, beta-phenylethyl alcohol, phenylacetaldehyde, *cis*-rose oxide, isobutyl acetate, and 2–3-pentanedione. This study revealed the changing laws of different thawing methods on the quality characteristics and flavor characteristics of Fo Tiao Qiang. It provides theoretical guidance for the industrial production and quality control of Fo Tiao Qiang.

## 1. Introduction

“Fo Tiao Qiang” occupies an important position in the traditional Chinese cuisine of Fujian, which is one of the provinces in China. It contains various popular seafoods and is rich in nutrients. Fo Tiao Qiang is usually made with abalone, sea cucumber, turtle skirt, isinglass, shiitake mushroom, tendon, flower mushroom, scallop, and pigeon egg as the ingredients, added to a boiled broth, and simmered. The soup stock ingredient of Fo Tiao Qiang includes fresh tube bones, Muscovy duck, pig intestine, native chicken, and fresh pigskin. The processing combines various cooking techniques such as frying, stir-frying, steaming, and simmering. Fo Tiao Qiang is popular among consumers because of its delicious mellow taste and high nutritional value. However, current research on Fo Tiao Qiang mainly focuses on cultural aspects, and scientific research mainly focuses on the aspect of the primary materials, sterilization technology, and shelf-life evaluation. In-depth studies on the nutritional quality and flavor characteristics of Fo Tiao Qiang are few.

Thawing is the reverse process of freezing and is one of the important factors affecting the quality of frozen products. The method of thawing frozen food has an important influence on the sensory, chemical, and microbiological quality of the food. Improper thawing methods can cause fat oxidation, protein denaturation, decreased water holding capacity and microbial contamination [1], resulting in the degradation of its quality. Recently, many new and efficient thawing methods are explored, such as ultrasonic thawing, microwave thawing, ohmic thawing [2], high-voltage electrostatic field thawing [3], and radio frequency [4]. Ultrasonication can speed up the thawing process and shorten thawing time [5]. An appropriate ultrasonic power (UAT-400 W) can effectively reduce the thawing loss of meat, helping to avoid mineral and water-soluble-vitamin loss during thawing [6]. Sun et al. [7] proposed that the cavitation effect of ultrasound also produces gas cores, which can effectively reduce mechanical damage to fish tissues, thereby reducing thawing losses. Microwave heating is done with an object that is heated rapidly because of the vibrations of water molecules in the microwaves electromagnetic field. It is superior to the other thawing methods based on heat transfer from the sample surface in terms of a short thawing time [8]. In addition, microwave thawing not only has the characteristics of fast heating efficiency, energy saving, safety, and easy control, but it also effectively maintains the nutritional quality and flavor of food [9].

Different thawing methods have different heat-transfer principles and different effects on product quality. Therefore, to improve the quality of thawed products, it is necessary to choose a more appropriate thawing method. In this study, Fo Tiao Qiang thawed through different thawing methods were analyzed by electronic tongue for free amino acids, taste activity value, and taste nucleotide, and by gas chromatography–ion mobility spectrometry (GC–IMS) for volatile flavors. They reveal the changing regular pattern of different thawing methods on the quality characteristics and flavor characteristics of Fo Tiao Qiang, as well as provide theoretical guidance for its industrial production and quality control.

## 2. Materials and Methods

### 2.1. Sample Preparation

All Fo Tiao Qiang samples were provided by Fujian Fliport Foods Co., Ltd. (Fuzhou, China), which were produced in January 2020. The ingredients and soup stock ingredients of Fo Tiao Qiang were burned in a porcelain altar for 8 ± 0.5 h with an open flame. After, the resultant product was canned, sterilized, cooled, and packaged.

The samples of Fo Tiao Qiang were taken out of the −18 °C freezer and randomly divided into four groups. We used the following methods to thaw until the soup melted as the end of the thawing.

(1)Ultrasonic thawing: The Fo Tiao Qiang sample was placed into the ultrasonic device with a power of 200 W, frequency of 40 kHz, and temperature of 25 ± 1 °C for thawing.(2)Microwave thawing: The Fo Tiao Qiang sample was placed into a microwave tray and microwaved at a frequency of 500 W to defrost.(3)Water bath thawing: The Fo Tiao Qiang sample was placed in a water bath to thaw at a temperature of 25 ± 1 °C.(4)Control group (natural thawing): The Fo Tiao Qiang samples were thawed in a ventilated room at a temperature of 25 ± 1 °C in a natural environment.

Research manuscripts reporting large datasets that are deposited in a publicly available database should specify where the data have been deposited and provide the relevant accession numbers. If the accession numbers have not yet been obtained at the time of submission, please state that they will be provided during review. They must be provided prior to publication.

Interventionary studies involving animals or humans, and other studies that require ethical approval, must list the authority that provided approval and the corresponding ethical approval code.

### 2.2. GC–IMS Analysis

Analysis of the volatile compounds in the Fo Tiao Qiang samples was performed using a GC–IMS instrument (G.A.S., Dortmund, Germany) with an FS-SE-54-CB-1 capillary column (15 m × 0.53 mm × 0.5 μm; Restek, Westport, CT, USA). The samples of Fo Tiao Qiang were analyzed with the GC–IMS instrument as described by Li et.al. [9] with slight modifications. One gram of Fo Tiao Qiang sample was accurately weighed and placed into a 20 mL headspace vial. The samples were then incubated for 15 min at 80 °C. After incubation, 500 μL of the HS gas was injected with a syringe into the injector at 85 °C. The rotating speed for incubating was 500 rpm.

Qualitative analysis of characteristic flavor substances was done by using Laboratory Analytical Viewer (LAV) and GC–IMS Library Search with the database of NIST 2014 (National Institute of Standards and Technology, Gaithersburg, MD, USA) and IMS2019 (G.A.S., Dortmund, Germany.). Reporter plug-in can directly compare the spectral differences between samples. Gallery plot plug-in can perform fingerprint comparison, as well as intuitive, quantitative comparison of the volatile organic compounds of samples. The dynamic principal component analysis (PCA) plug-in is used to cluster samples and quickly determine the types of unknown samples.

### 2.3. Electronic-Tongue Analysis

The samples of Fo Tiao Qiang were analyzed with the electronic tongue as described by Ren et.al. [10] with slight modifications. The samples were first thawed in a hot water bath water (about 40 °C). The test on machine was done by mixing the sample (20 g) and purified water (80 g, Wahaha purified water) in a 250 mL beaker. The taste perception mechanism of living animals was imitated by using a taste analysis system (type of TS-5000Z) and the lipid membrane of an artificial sensor with wide area selection specificity. Using the electronic tongue, we detected the changes in the membrane potential caused by electrostatic or hydrophobic interactions between various flavor substances and artificial lipid membranes to evaluate the five basic flavors and the astringency. Artificial saliva mixed with KCl (30 mM) and tartaric acid (0.3 mM) was used as the reference solution. Each sample was tested three times in parallel, the data were analyzed using the device’s own database and software, and the radar chart and bubble chart were drawn.

### 2.4. Free Amino Acid (FAA) Component Analysis

The FAA component was determined using the AQC(6-Aminoquinolyl-N-hydroxysuccinimidyl Carbamate) derivative method described by Zhou et al. [11] with slight modifications. Approximately 10.0 μL of the supernatant of the centrifugal fluid was filtered through a 0.22 μm microporous membrane and mixed with 70.0 μL of AccQ-Fluor borate buffer (pH 8.8), after which 20.0 μL of AccQ-Fluor reagent (3000 mg/L) was added. The solution was then vortexed and placed in an oven for 10 min at 55 °C. Finally, the solution was cooled to room temperature for testing.

### 2.5. Taste Activity Value (TAV) Analysis

The TAV was calculated according to the following formula [12].
TAV = absolute concentration of a certain odorous substance in the sample/taste threshold of the substance

### 2.6. Measurement of Protein, Fat, Hydroxyproline, Total Sugar Content

Protein content was determined by the Kjeldahl method according to China National Standard: GB 5009.5-2016. Fat content was determined by acid hydrolysis according to China National Standard: GB 5009.6-2016. Hydroxyproline was determined by the colorimetric evaluation according to China National Standard: GB T9695.23-2008. Total sugar content was determined by spectrophotometry according to China National Standard: GB T9695.31-2008. 

### 2.7. Taste Nucleotide

TAV was determined by liquid chromatography according to China National Standard: GB 5413.40-2016.

### 2.8. Equivalent Umami Concentration (EUC)

The EUC was calculated according to the following formula [13]:(1)$$\;\mathrm{EUC}\left(g\mathrm{MSG}/100\;g\right)=\sum a_{i}b_{i}+1218\left(\sum a_{i}b_{i}\right)\left(\sum a_{i}b_{i}\right),$$
where *a_i_* is amount of umami amino acids (g/100 g), and *b_i_* is the umami coefficient of umami amino acids relative to monosodium glutamate (MSG) (glutamate is 1.0; aspartic acid is 0.077). *a_j_* is the quantity of taste nucleotide/(g/100 g), *b_j_* is the umami coefficient of taste nucleotide relative to IMP (IMP(disodiuminosine5’-monophosphate) is 1.0; AMP(adenosine monophosphate) is 0.18; GMP(disodium guanosine5’-monophosphate) is 2.3), and 1218 is the synergy constant.

### 2.9. Statistical Analysis

Each sample was measured three times for repeated experiments. The resulting data are represented as mean ± S.D. The experimental data were determined by ANOVA with SPSS 25.0 software (SPSS Inc., Chicago, IL, USA). The table was generated using Excel 2016 software (Microsoft Corporation, Redmond, WA, USA). The radar charts, bubble charts, and multiple histograms were generated using Origin 2018 (OriginLab, Massachusetts, MA, USA).

## 3. Results

### 3.1. Protein, Fat, Hydroxyproline, and Total Sugar Content

The protein, fat, hydroxyproline, and total sugar content of different thawing methods of Fo Tiao Qiang are shown in Table 1. There was no significant difference in the protein, hydroxyproline, and total sugar content of the four samples of Fo Tiao Qiang. The content of hydroxyproline and total sugar in the four thawing methods was not much different. The protein content of the samples from ultrasonic thawing were slightly higher than other thawing methods. One study showed that ultrasonic thawing can contribute to the formation of smaller particle sizes and a higher solubility, which promote an increase in protein solubility and the formation of soluble protein [14]. In addition, the cavitation effect not only destroys the hydrophobic interactions that lead to the cross-linking of protein aggregates [15], but it also expands the structure of myofibrillar protein (MP) in combination with mild protein oxidation, thereby further promoting the increase in solubility [16]. Therefore, the protein content of the sample thawed by ultrasound is slightly higher than that of the other three thawing methods. Among the four samples from the different thawing methods, the crude fat content of the microwave-thawed Fo Tiao Qiang sample was significantly lower than that of the control group. This is mainly due to the increase in temperature, which caused the oxidation and decomposition of fat. According to reports, lipid oxidation is more likely to occur in regions with higher temperatures and pressures, and generates more free radicals with strong oxidizing abilities [17].

### 3.2. FAA Component

FAAs are important taste components. Table 2 shows the amino acid composition and the corresponding taste activity value of the samples from different thawing methods. A total of 17 FAAs including eight essential amino acids and nine non-essential amino acids were identified. From the result, the perspective of the total amount of amino acids under the four different thawing methods is significantly different (*p* < 0.05). The total amounts of amino acids from natural thawing, ultrasonic thawing, microwave thawing, and water bath thawing were 1224.40, 1290.79, 1217.67, and 1270.79 mg/100 g, respectively. A prolonged thawing time at a high temperature or local overheating will accelerate protein denaturation and lead to a significant reduction in free amino groups [14]. Studies have shown that ultrasonic thawing has a better stability and causes less damage to food [7]. Hence, the total amount of amino acids thawed by ultrasound was the highest, which is similar to the results of Bou et al. [18]. Zhang et al. [19] summarized three possible reasons that might be involved in the changes of FAAs. (1) The MP structure could be destroyed by ultrasonic treatment and thus more FAAs would migrate from the samples. (2) FAAs could go through thermal degradation under a high temperature and react with the reduced sugar, leading to the Maillard reaction. (3) FAAs could actively participate in Strecker degradation and further produce many volatile flavor compounds. Therefore, the amino acid content in different thawing processes is related to the amount of protein and amino acid degradation. 

In this study, the higher FAA content of the Fo Tiao Qiang after thawing was from glutamic acid, alanine, and arginine. Glutamic acid and alanine are important fresh and sweet amino acids in the Fo Tiao Qiang. Among all amino acids, the four thawing methods had the highest glutamic acid content. The ranking was ultrasonic thawing > water bath thawing > natural thawing > microwave thawing, which was consistent with the total amino acid ranking, indicating that ultrasonic thawing is beneficial to the release of amino acids [20]. 

TAV is proportional to the taste intensity. The synergistic interaction of various taste compounds is probably the most important factor affecting the taste of meat products [19]. It can be seen from the table that the amino acids that contributed much to the taste in the natural-thawing sample were glutamic acid (fresh), arginine (bitter/sweet), and alanine (sweet). The amino acids that contributed greatly to the taste in the ultrasonic-thawing sample were glutamic acid (fresh), methionine (bitter), and histidine (bitter). The amino acids that contributed the most to the taste in the microwave-thawing sample in decreasing order were glutamic acid (fresh), arginine (bitter/sweet), and alanine (sweet). In the water-bath-thawed sample, the most important contributions to the taste were glutamic acid (fresh), arginine (bitter/sweet), and alanine (sweet), the same as for natural thawing and microwave thawing. It was further proved that glutamic acid is the most important flavor amino acid in Fo Tiao Qiang after thawing.

### 3.3. Taste Nucleotide

The content of taste nucleotides and their differences in different thawing methods are shown in Figure 1. Except for GMP, the other three nucleotides have significant differences. 

The contents of 5′-IMP, 5′-GMP, and 5′-AMP of different thawing methods and their taste activity values are shown in Table 3. Among the three kinds of nucleotides (Table 3), IMP has the highest content, indicating that IMP is the most active nucleotide in the Fo Tiao Qiang. Studies have shown that IMP is considered as a flavor enhancer and is widely used in the food industry to improve taste [21]. The synergistic effect between IMP and GMP can strongly enhance freshness [22]. Among the four thawing methods, the AMP and GMP values of ultrasonic thawing were the highest, but the IMP value was lower than that of microwave thawing. This may be because IMP has thermal instability, and the cavitation effect during ultrasonic thawing produces high temperatures and high pressures, which leads to thermal degradation and the loss of 5′-IMP [21,23].

The EUC (MSG equivalent) is generally used to measure the synergy between flavored nucleotides and savory amino acids. Research by Sabikun et al. [24] showed that 5′-nucleotides (IMP, GMP, and AMP) have a synergistic effect with aspartic and glutamic acids. After calculation, the EUC of the four thawing methods are in the order of 48.72, 60.73, 48.61, and 58.18 g MSG/100 g. The results show that the Fo Tiao Qiang with ultrasonic thawing has the highest umami intensity.

### 3.4. Electronic Tongue

#### 3.4.1. Flavor Profile

Figure 2 shows the outline of the Fo Tiao Qiang in different ways of thawing. We take the output of the reference solution as the tasteless point (tasteless, or 0 point), the tasteless point for the sour taste is −13, for the salty taste is −6, and for the other indicators is 0. According to this, when the taste value of the sample is lower than tasteless, it means that the sample does not have a taste; otherwise, it has taste. It can be seen from the radar chart of taste indicators that the Fo Tiao Qiang samples with different thawing methods have no sour taste. Compared with the reference solution, samples with different thawing methods have a certain difference in taste. As shown in the figure below, there are obvious differences in umami, saltiness, bitterness, and richness among the four thawing methods.

#### 3.4.2. Umami, Saltiness, and Richness

Figure 3 is the bubble chart of different thawing methods of the Fo Tiao Qiang, including saltiness, umami, and richness. The differences in saltiness, umami, and richness of the Fo Tiao Qiang should be specifically analyzed with different thawing methods. As shown in Figure 3, the stability of the three parallels of the same sample is better. The richness (the size of the bubbles) is compared first. The bigger the bubble, the greater the richness. The flavor richness of the Fo Tiao Qiang samples obtained by ultrasonic, microwave, and water-bath thawing was significantly higher than that of natural thawing. Second, the saltiness (X-axis) was compared. The saltiness of the samples obtained by water-bath thawing was significantly reduced and lower than for natural thawing. 

The saltiness of the samples obtained by microwave thawing was closer to natural thawing, and the saltiness of the ultrasonic-thawing samples was slightly higher than for natural thawing and water-bath thawing. Lastly, the difference in umami (Y axis) was analyzed. Compared with natural thawing, the umami taste of the samples with the water bath thawing decreased, and the umami of the samples with the microwave and ultrasonic thawing increased. Moreover, the highest umami value was the sample obtained by ultrasonic thawing. This was consistent with EUC (monosodium glutamate equivalent) findings.

#### 3.4.3. Bitterness, Astringency, and Bitter Aftertaste

Figure 4 is the bubble chart of different thawing methods of the Fo Tiao Qiang, including bitterness, astringency, and bitterness aftertaste. The bitterness and astringency of the Fo Tiao Qiang obtained by different thawing methods are analyzed as follows. As shown in Figure 4, the samples obtained by microwave thawing were close in terms of bitterness to the samples with ultrasonic thawing. The bitterness and bitter aftertaste of the samples in ultrasonic thawing were slightly enhanced compared with microwave thawing. Compared with natural thawing, the astringency of samples obtained from the other three thawing methods were significantly decreased, especially for water-bath thawing. 

#### 3.4.4. Principal Component Analysis (PCA)

The results of PCA on the ingredient data are shown in Figure 5, in which samples with different thawing methods are represented by points of different colors. According to the PCA, the variance contribution rates of the first principal component (PC1) and the second principal component (PC2) were 92.3% and 7.0%, respectively, and the total contribution rate was 99.3%. The results show that PC1 and PC2 already contained a very large amount of information of the sample, which represented the original information of the sensor and reflected the overall information of the sample. The results in the figure show that there was a small overlap in the PCA distribution between the ultrasonic-thawing samples and the microwave-thawing samples, and so their overall tastes were relatively similar. There was an obvious difference between the natural thawing and water-bath-thawing sample, and the natural thawing was more significant.

### 3.5. Volatile Substances

#### 3.5.1. Comparative Analysis of GC–IMS Spectra

Figure 6 is the GC–IMS spectra of different thawing methods generated by the Reporter plug-in program in the LAV analysis software. The data were visually represented using a 3D spectrogram. In Figure 1, the *x*-, and *y*-axes, respectively, represent the ion migration time for identification (Dt) and the retention time of the gas chromatograph (Rt). In the figure, each point represents a volatile organic compound. From Figure 5 and Figure 6, the characteristic volatile components of the samples with different thawing methods have different GC–IMS characteristic spectrum information.

#### 3.5.2. Qualitative Volatile Components

Figure 7 shows the GC–IMS 2D spectra from Fo Tiao Qiang from different thawing methods. The entire spectra represented all volatile compounds of the samples. The red vertical line on the left side indicates the reactive ion peak (RIP), and each point on both sides of the RIP represents a volatile organic compound from the samples. The color represents the signal strength of the substance. White indicates a lower intensity, and red indicates a higher intensity. The intensity increased as the color deepened. From Figure 7, it can be intuitively concluded that there were fewer types of volatile compounds in natural thawing and water-bath thawing. The NIST database and IMS database built in the GC–IMS Library Search software was used to qualitatively analyze the substances. A total of 42 monomers and dimers of some substances were qualitatively detected, such as alcohols, aldehydes, ketones, esters, and other categories. 

Figure 8 shows the fingerprints of volatile substances in the different thawing methods. In the fingerprint, each row represents all volatile organic substances detected in the sample, and each column is a comparison of the same substance between different samples. We can see in Figure 8 that flavor substances such as alpha-terpineol, beta-phenylethyl alcohol, phenylacetaldehyde, *cis*-rose oxide, isobutyl acetate, and 2–3-pentanedione in area A were thawed by microwave, ultrasonic thawing, and water-bath thawing. Compared with the control sample, contents of these flavor substances were significantly increased. It may be because microwave thawing, ultrasonic thawing, and water-bath thawing promote molecular movement and aggravate the oxidation of fat. Terpineol, phenethyl alcohol, and rose ether have floral scents. The content of phenylacetaldehyde and *cis*-rose oxide (phenylacetaldehyde, *cis*-rose ether) in the water-bath-thawed samples increased relatively little. Studies have shown that ultrasonic treatment can accelerate lipid oxidation to produce more volatile flavor compounds, such as aldehydes, ketones, etc., and improve flavor by enhancing the interaction between volatile compounds and meat proteins [18,21,25].

In area B, a large amount of flavor substances exist in the control sample, such as aldehydes (pentanal, hexanal, heptanal, (*E*)-2-heptenal, octanal, (*E*)-2-octenal, 3-methylbutanal, 2-furfural, and benzaldehyde), alcohols (maltol, 3-methylbutan-1-ol, and 5-methylfurfuryl alcohol), and ketones (2-butanone, 1-octen-3-one). The content of the microwave-thawed sample was slightly less than that of the control sample, and the content of the water-bath-thawed sample was the least. The main reasons are that the control group had not been heated for a long time, the tissue destruction rate was low, and the protein was not easily denatured. 3-Methylbutan-1-ol, 2-furfural, pentanal, and other substances had also disappeared in the water-bath thawed samples.

In area C, the water-bath-thawing sample contents of 6-methyl-5-hepten-2-one, 3-hydroxy-2-butanone, 2-heptanone, ethyl acetate, isovaleric acid, 1-octen-3-ol, (*Z*)-3-hexenol, alpha-pinene, and other substances were greatly increased. Isovaleric acid is a food flavor permitted by GB 2760-1996. It is mainly used to prepare cheese and cream flavors. It is also used in a trace amount of fruit flavors. Ethyl acetate has a fruity aroma, and the test data of amino acids indicates that it can be produced by more flavored amino acids by thawing in a static water bath.

In summary, natural thawing, microwave thawing, ultrasonic thawing, and water-bath thawing reduced the content of most flavor substances. Here, water-bath-thawed samples declined the most. It may be because the materials in the tank were not uniformly heated by static water bath heating, and the low temperature inhibited the oxidation and decomposition of fat to a certain extent. The content of some flavor substances showed different degrees of improvement in samples obtained by microwave thawing, ultrasonic thawing, and water-bath thawing.

#### 3.5.3. GC–IMS PCA

The GC–IMS PCA results of the volatile flavor compounds of Fo Tiao Qiang under different thawing methods are shown in Figure 9. The contribution rate of PC1 was 56%, the contribution rate of PC2 was 28%, and the cumulative contribution rate was 84% (>70%), indicating that Figure 9 can represent the information of volatile substances of Fo Tiao Qiang. By observing Figure 9, we can see that the four thawing methods were clustered separately and did not overlap. The distance between microwave thawing and ultrasonic thawing was close, indicating that the volatile substances of the two samples were similar.

## 4. Summary

In this study, four methods (natural thawing, ultrasonic thawing, microwave thawing, and water-bath thawing) were used to thaw Fo Tiao Qiang.

From the perspective of physical and chemical indicators, the protein content of the ultrasonically thawed Fo Tiao Qiang is slightly higher than that of other thawing methods; the fat content of the microwave-thawed Fo Tiao Qiang is significantly lower than that of the other three types of Fo Tiao Qiang.

The FAA content is determined by the combination of amino acids produced by protein solubilization and the loss of amino acids caused by amino acid degradation. Ultrasonic thawing has a good stability, and so the sample thawed by ultrasonic has the highest amino acid content. Among the four thawing methods, the glutamic acid content is the highest, which contributes the most to the taste.

Taste nucleotides and umami amino acids have a synergistic effect, and EUC is generally used to measure the synergistic effect. Ultrasonically thawed Fo Tiao Qiang has the highest intensity of umami flavor, and the nucleotide that contributed the most to the flavor of Fo Tiao Qiang was 5′-IMP.

The electronic tongue technology intuitively reflects the obvious difference in the taste of Fo Tiao Qiang under different thawing methods. According to the PCA chart, the overall tastes from ultrasonic thawing and microwave thawing are similar. GC–IMS identified 42 compounds of Fo Tiao Qiang. Microwave thawing, ultrasonic thawing, and water-bath thawing led to a decrease in the content of most flavor substances and an increase in the content of some flavor substances.

The above results can provide a theoretical basis for the nutritional index and flavor changes in different thawing methods in order to find the best thawing method.

## Figures and Tables

**Figure 1 foods-11-01330-f001:**
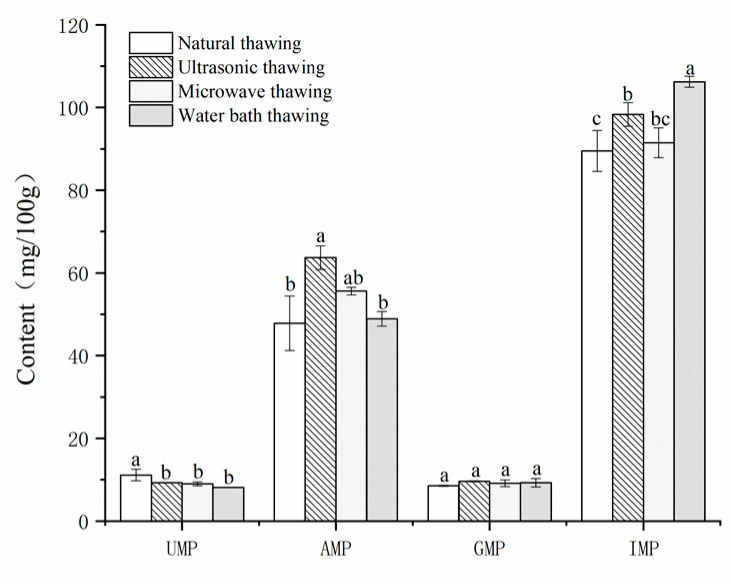
Nucleotide content of different thawing method “Fo Tiao Qiang”. Note: Different letters in the figure indicate significant differences (*p* < 0.05).

**Figure 2 foods-11-01330-f002:**
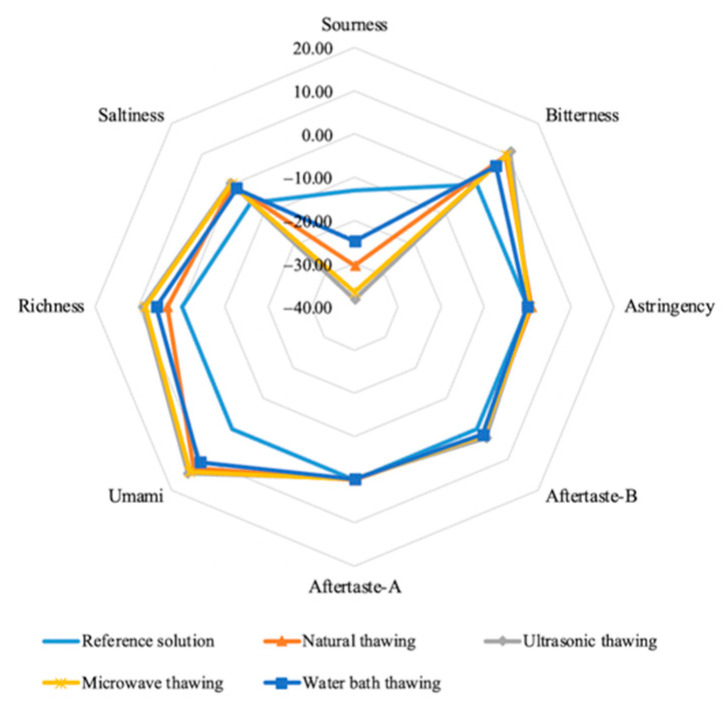
Radar chart of effective taste index of samples of different thawing methods of “Fo Tiao Qiang”.

**Figure 3 foods-11-01330-f003:**
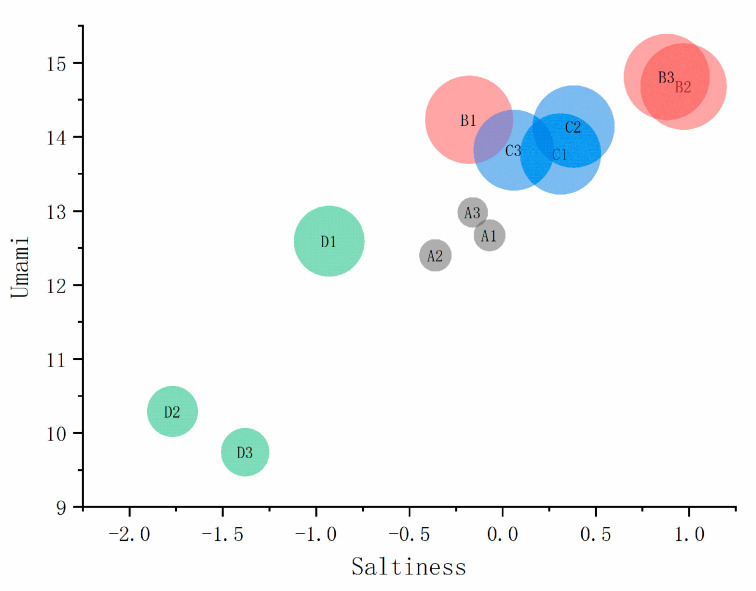
Saltiness, umami, and richness bubble chart of different thawing methods of *“Fo Tiao Qiang”*. Note: “A1A2A3”, “B1B2B3”, “C1C2C3”, “D1D2D3” are three parallel samples of “natural thawing, ultrasonic thawing, microwave thawing, and water bath thawing”.

**Figure 4 foods-11-01330-f004:**
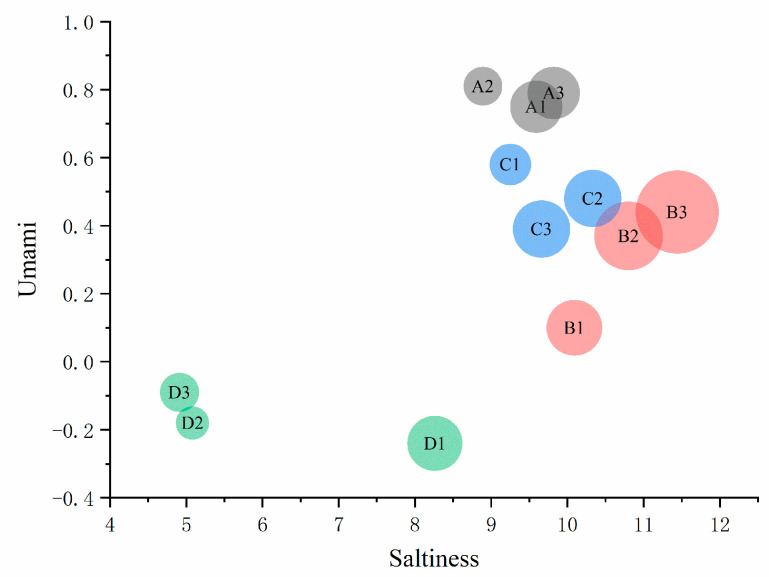
Bitterness, astringency, and bitterness aftertaste bubble chart of different thawing methods of “Fo Tiao Qiang”. Note: “A1A2A3”, “B1B2B3”, “C1C2C3”, “D1D2D3” are three parallel samples of “natural thawing, ultrasonic thawing, microwave thawing, and water bath thawing”.

**Figure 5 foods-11-01330-f005:**
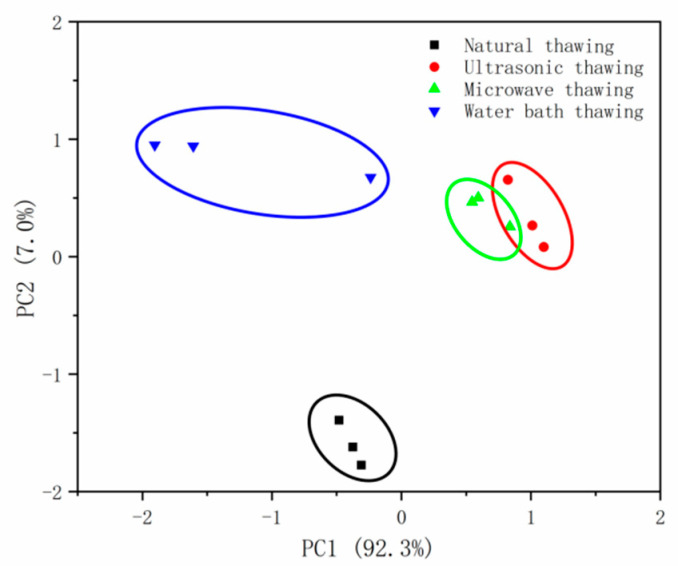
PCA principal component analysis diagram of different thawing methods of “Fo Tiao Qiang”.

**Figure 6 foods-11-01330-f006:**
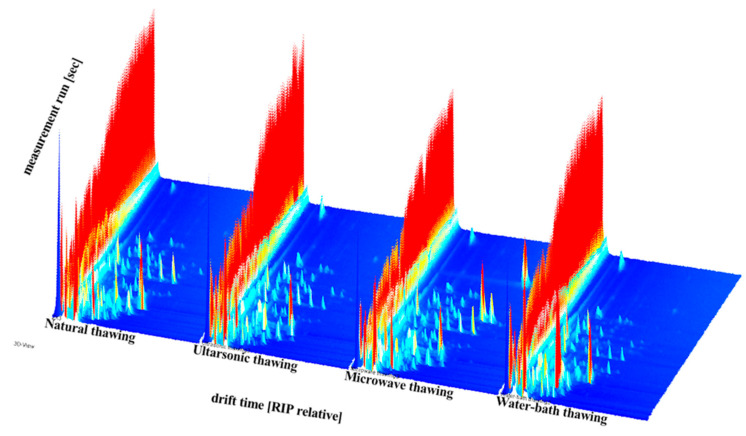
3D-topographic of different thawing methods of “Fo Tiao Qiang”.

**Figure 7 foods-11-01330-f007:**
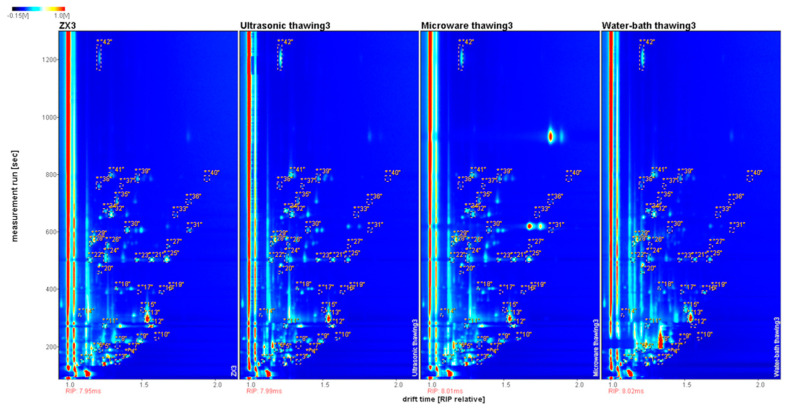
Results of GC–IMS spectra in different thawing methods of “Fo Tiao Qiang”.

**Figure 8 foods-11-01330-f008:**
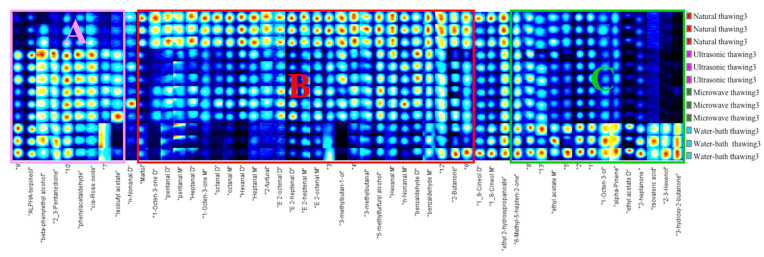
GC–IMS volatile substances fingerprint in different thawing methods of “Fo Tiao Qiang”. Note: A, B, C respectively represent the characteristic flavor substances of Fo Tiao Qiang under different thawing methods.

**Figure 9 foods-11-01330-f009:**
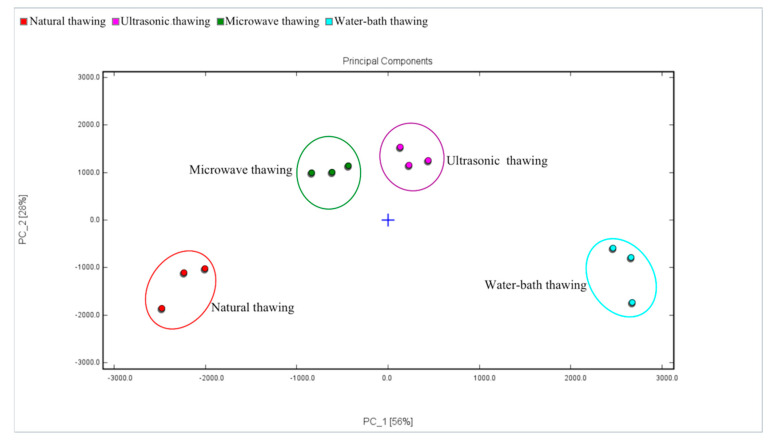
PCA analysis chart of different thawing method of “Fo Tiao Qiang”.

**Table 1 foods-11-01330-t001:** The protein, hydroxyproline, fat, and total sugar content of different thawing methods of “Fo Tiao Qiang”.

Sample	Protein (g/100 g)	Hydroxyproline (g/100 g)	Fat (g/100 g)	Total Sugar (g/100 g)
Natural thawing	75.57 ± 1.57 ^a^	0.016 ± 0.00 ^a^	0.249 ± 0.02 ^a^	0.0162 ± 0.00 ^a^
Ultrasonic thawing	76.06 ± 4.18 ^a^	0.016 ± 0.00 ^a^	0.197 ± 0.03 ^ab^	0.0153 ± 0.00 ^a^
Microwave thawing	74.21 ± 1.67 ^a^	0.014 ± 0.00 ^a^	0.143 ± 0.00 ^b^	0.0162 ± 0.00 ^a^
Water bath thawing	74.35 ± 4.12 ^a^	0.015 ± 0.00 ^a^	0.175 ± 0.09 ^ab^	0.0154 ± 0.00 ^a^

Note: Values are expressed as mean ± SD (n = 3), and different letters in the column indicate significant differences (*p* < 0.05).

**Table 2 foods-11-01330-t002:** Composition of amino acids in different thawing methods of “Fo Tiao Qiang”.

Amino Acid	Taste Contribution	Threshold (mg/100 g)	Natural Thawing		Ultrasonic Thawing		Microwave Thawing		Water bath Thawing	
			Content (mg/100 g)	TAV	Content (mg/100 g)	TAV	Content (mg/100 g)	TAV	Content (mg/100 g)	TAV
Asp	fresh	100	58.90 ± 3.88 ^b^	0.59	69.34 ± 1.97 ^a^	0.69	61.18 ± 0.74 ^ab^	0.61	69.20 ± 7.71 ^a^	0.69
Glu	fresh	30	332.84 ± 8.20 ^b^	11.09	371.00 ± 3.42 ^a^	11.43	318.91 ± 4.43 ^c^	10.63	342.84 ± 3.12 ^b^	11.43
Ger	sweet	150	62.25 ± 3.86 ^a^	0.41	72.52 ± 2.59 ^a^	0.41	64.80 ± 2.12 ^a^	0.43	62.24 ± 8.01 ^a^	0.41
Gly	sweet	130	104.06 ± 3.76 ^b^	0.80	96.74 ± 0.85 ^b^	0.48	121.61 ± 4.08 ^a^	0.94	130.12 ± 6.2 ^a^	1.00
Thr	sweet	260	42.67 ± 1.31 ^a^	0.16	44.77 ± 0.20 ^a^	0.50	41.82 ± 0.27 ^a^	0.16	42.66 ± 3.21 ^a^	0.16
Ala	sweet	60	108.89 ± 0.93 ^c^	1.81	119.26 ± 2.68 ^a^	0.71	110.57 ± 4.36 ^bc^	1.84	116.01 ± 2.18 ^ab^	1.93
Pro	sweet	-	66.05 ± 1.31 ^b^	/	68.50 ± 0.73 ^ab^	/	66.20 ± 0.51 ^b^	/	71.76 ± 4.37 ^a^	/
Lys	sweet/bitter	50	62.49 ± 1.84 ^a^	1.25	63.95 ± 0.20 ^a^	1.44	57.81 ± 0.64 ^b^	1.16	62.84 ± 3.67 ^a^	1.26
Asn	sweet	-	44.96 ± 0.82 ^a^	/	49.76 ± 2.66 ^a^	/	46.28 ± 2.29 ^a^	/	45.91 ± 3.24 ^a^	/
Tyr	bitter	-	41.45 ± 1.11 ^b^	/	46.93 ± 1.97 ^a^	/	41.09 ± 0.51 ^b^	/	41.51 ± 1.77 ^b^	/
Val	bitter/sweet	40	47.96 ± 0.41 ^b^	1.20	50.98 ± 2.96 ^a^	1.04	47.24 ± 0.28 ^b^	1.18	50.80 ± 1.31 ^a^	1.27
Iel	bitter	90	33.70 ± 2.50 ^a^	0.37	28.98 ± 2.20 ^b^	0.56	29.43 ± 2.62 ^ab^	0.33	27.96 ± 0.69 ^b^	0.31
Leu	bitter	190	67.08 ± 1.15 ^a^	0.35	64.05 ± 1.84 ^ab^	0.00	61.23 ± 1.52 ^b^	0.32	61.31 ± 2.51 ^b^	0.32
Arg	bitter/sweet	50	105.41 ± 5.86 ^a^	2.11	95.37 ± 5.91 ^a^	0.56	104.59 ± 0.06 ^a^	2.09	100.62 ± 1.0 ^a^	2.01
His	bitter	20	26.69 ± 0.10 ^b^	1.33	28.68 ± 0.19 ^a^	3.07	26.02 ± 0.52 ^b^	1.30	28.58 ± 0.56 ^a^	1.43
Met	bitter	30	12.03 ± 0.28 ^bc^	0.40	14.27 ± 0.62 ^a^	3.35	29.43 ± 2.62 ^ab^	0.33	11.17 ± 1.23 ^c^	0.37
Trp	bitter	-	6.98 ± 0.23 ^a^	/	5.71 ± 0.29 ^b^	/	13.55 ± 0.20 ^ab^	/	4.67 ± 0.95 ^b^	/
Totol			1224.40 ± 37.55 ^b^		1290.79 ± 31.28 ^a^		1217.67 ± 25.22 ^b^		1270.19 ± 53.98 ^ab^	

Note: TAV: Taste Activity Value. “-” did not find data; “/” was not calculated; the same line marked different letters indicates significant differences (*p* < 0.05).

**Table 3 foods-11-01330-t003:** Nucleotide content and their TAVs of different thawing method “Fo Tiao Qiang”.

Nucleotide	Threshold (mg/100 g)	Natural Thawing		Ultrasonic Thawing		Microwave Thawing		Water Bath Thawing	
		Content (mg/100 g)	TAV	Content (mg/100 g)	TAV	Content (mg/100 g)	TAV	Content (mg/100 g)	TAV
AMP Adenosine	50	47.82 ± 6.58 ^b^	0.96	63.73 ± 2.87 ^a^	1.27	55.62 ± 0.91 ^ab^	1.11	48.90 ± 1.76 ^b^	0.98
GMP Guanylic acid	12.5	8.52 ± 0.14 ^a^	0.68	9.60 ± 0.14 ^a^	0.77	9.14 ± 0.80 ^a^	0.73	9.28 ± 0.99 ^a^	0.74
IMP Inosinic acid	25	89.53 ± 4.95 ^c^	3.58	98.34 ± 2.84 ^b^	3.93	91.48 ± 3.59 ^b^	3.66	106.23 ± 1.33 ^a^	4.25

Note: the same line marked different letters indicates significant differences (*p* < 0.05).

## Data Availability

Data is contained within the article.

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
