# Peer review of "Study on the Flavor Compounds of Fo Tiao Qiang under Different Thawing Methods Based on GC–IMS and Electronic Tongue Technology"

_foods, 2022, doi:10.3390/foods11091330_

Round 1
Reviewer 1 Report
I have reviewed the manuscript entitled "Study on the flavor compounds of Fo Tiao Qiang under different thawing methods based on GC-IMS and electronic tongue technology” by Lin et al. The manuscript investigates the free amino acids, taste activity value, and taste nucleotide and volatile flavors in Fo Tiao Qiang obtained from different thawing methods. The paper is well presented and easy to read. The introduction provides a good, generalized background of the topic that quickly gives the reader an appreciation of the wide range of applications for flavour science. The literature cited is relevant to the study. However, there are some parts that require revision for better understanding of the readers. I think the paper could prove to be very interesting and useful to very large researchers.
My remarks about the text are as follows:
2.1. Sample preparation
Add the providing year of the Fo Tiao Qiang samples.
Thawing conditions (watt, temperature, time etc) of the samples are determined according to what criteria? Explain.
2.5. Taste activity value (TAV) analysis: Add a reference for this section.
2.6. Measurement of protein, fat, hydroxyproline, total sugar content: Add a reference for this section.
2.7. Taste nucleotide: explain this analysis.
Improve the resolution of Figures.
Reviewer 2 Report
After reviewing a manuscript "Study on the flavor compounds of Fo Tiao Qiang under different thawing methods based on GC-IMS and electronic tongue technology", I suggest the following corrections:
- Line 17: …. "the protein content was higher in Fo Tiao Qiang with"…It should be written that protein content was slightly higher; as it was written in the results section, line 167.
- Lines 135-139: Provide references for the applied methods: Kjeldahl method; fat, hydroxyproline and sugar content determination. Which method was used for sugar content determination (phenol-sulfuric acid, anthrone test, etc…)
- Line 141: Chromatography method for the determination of taste nucleotide was not described and reference is lacking.
- Section 3.1. Protein, fat, hydroxyproline, and total sugar content: discussion for the sugar content is lacking.
- Lines 388-389: it should be written that protein content is slightly higher…
- Lines 389-390: ……."but the sugar content is lower"….. Check the statement. Sugar content is not lower according to the significant difference calculations, Table 1
